# Perspectives on the Use of a Medium-Dose Etoposide, Cyclophosphamide, and Total Body Irradiation Conditioning Regimen in Allogeneic Hematopoietic Stem Cell Transplantation: The Japanese Experience from 1993 to Present

**DOI:** 10.3390/jcm8050569

**Published:** 2019-04-26

**Authors:** Masahiro Imamura, Akio Shigematsu

**Affiliations:** 1Former Professor of Department of Hematology, Hokkaido University Faculty of Medicine, Graduate School of Medicine, Kita-15 Jyo, Nishi-7 Chome, Kita-ku, Sapporo 060-8638, Japan; 2Department of Hematology, Sapporo Hokuyu Hospital, 6-6-5-1 Higashi-sapporo, Shiroishi-ku, Sapporo 003-0006, Japan; shigemap@mac.com

**Keywords:** allogeneic hematopoietic stem cell transplantation, acute lymphoblastic leukemia, conditioning regimen, medium-dose etoposide, cyclophosphamide, total body irradiation, pharmacokinetics, pharmacodynamics

## Abstract

The outcome for adults with acute lymphoblastic leukemia (ALL) treated with chemotherapy or autologous hematopoietic stem cell transplantation (HSCT) is poor. Therefore, allogeneic HSCT (allo HSCT) for adults aged less than 50 years with ALL is performed with myeloablative conditioning (MAC) regimens. Among the several MAC regimens, a conditioning regimen of 120 mg/kg (60mg/kg for two days) cyclophosphamide (CY) and 12 gray fractionated (12 gray in six fractions for three days) total body irradiation (TBI) is commonly used, resulting in a long term survival rate of approximately 50% when transplanted at the first complete remission. The addition of 30 mg/kg (15 mg/kg for two days) etoposide (ETP) to the CY/TBI regimen revealed an excellent outcome (a long-term survival rate of approximately 80%) in adults with ALL, showing lower relapse and non-relapse mortality rates. It is preferable to perform allo HSCT with a medium-dose ETP/CY/TBI conditioning regimen at the first complete remission in high-risk ALL patients and at the second complete remission (in addition to the first complete remission) in standard-risk ALL patients. The ETP dose and administration schedule are important factors for reducing the relapse and non-relapse mortality rates, preserving a better outcome. The pharmacological study suggests that the prolonged administration of ETP at a reduced dose is a promising treatment.

## 1. Introduction

The outcome for adults with acute lymphoblastic leukemia (ALL) treated with chemotherapy or autologous hematopoietic stem cell transplantation is poor [1,2,3], given that the long-term survival rate is approximately 30% with a high incidence of relapse; therefore, allogeneic hematopoietic stem cell transplantation (allo HSCT) is utilized as an essential treatment modality for such patients. Nevertheless, an unsatisfactory long-term survival rate of approximately 50% has been shown in adults with acute lymphoblastic leukemia (ALL) who underwent allo HSCT at the first complete remission (CR1), and worse outcomes have been shown in more advanced stages (i.e., CR2, beyond CR2 or non-CR) [3,4,5,6,7,8]. To improve the outcome of allo HSCT in adults with ALL, there exist several points that should be efficiently managed; namely, control of acute and chronic graft-versus-host disease (GVHD), enhancement of graft-versus-leukemia effect, and reduction of relapse and non-relapse mortality (NRM). In preventing relapse, the selection of a conditioning regimen is one of the important factors. Although various reduced intensity conditioning (RIC) regimens were introduced in the late 1990s and older adults with ALL have been treated with such regimens [9,10,11,12,13], myeloablative conditioning (MAC) regimens are still generally used and several intensified conditioning regimens have been developed [14,15,16,17,18,19,20,21]. MAC regimens of 120 mg/kg (60mg/kg for two days) cyclophosphamide (CY) and 12 to 13.2 gray (Gy) fractionated (six fractions for three days) total body irradiation (TBI) have been commonly applied to allo HSCT [22,23]. Other MAC regimens such as the busulfan-based [15,17] or etoposide (ETP)-based [14,15,16,17,18,19,20] regimens had also been used. Although the ETP-based regimens (usually 40–70 mg/kg as a single dose) showed efficacy to some extent, severe toxicities were often seen [24]. Among those regimens, the medium-dose (intravenous administration over three hours at a dose of 15 mg/kg once daily for two days) ETP added to a 120 mg/kg CY plus 12 Gy TBI conditioning regimen appears to be promising for allo HSCT in adults aged under 50 years with ALL when transplanted at CR1 and also at CR2, showing an excellent outcome without increasing relapse and transplant-related mortality (TRM) rates [25,26,27,28,29]. In contrast, RIC regimens have also been applied to adults with ALL and favorable outcomes have been obtained; however, relapse and TRM rates remain high [9,10,11,12,13]. Therefore, an allo HSCT conditioning regimen which deserves further study for adult ALL patients aged less than around 50 years at CR1 and CR2 appears to be the medium-dose ETP/CY/TBI and RIC is suitable for patients aged over 55 years [30] or for younger patients with comorbid conditions. On the contrary, new therapeutic strategies such as various tyrosine kinase inhibitors for Philadelphia chromosome (Ph)-positive ALL [31,32,33,34], human leukocyte antigen-haploidentical HSCT [35,36,37,38], pediatric-inspired regimens for Ph-negative ALL [39,40], chimeric antigen receptor T cell therapy [41,42,43], and immunotherapy with monoclonal antibodies [44,45,46,47] for patients with ALL are increasingly utilized with favorable outcomes. Therefore, we are required to select an appropriate treatment modality, considering each patient’s characteristics. Although we have various treatment modalities for adults with ALL, allo HSCT with MAC is still one of the favorable treatments. In this manuscript, we present the beneficial effect of a medium-dose ETP/CY/TBI conditioning regimen for adults with ALL who underwent allo HSCT at CR based on retrospective analysis mainly performed at the Hokkaido University Hospital, where a reduced-dose ETP/CY/TBI conditioning regimen has been utilized in allo HSCT for adults with various hematological malignancies since 1990, a retrospective analysis using a Japanese transplant registry database and a prospective multi-center phase II clinical trial in Japan. Furthermore, we discuss how we can further improve the potent efficacy of this conditioning regimen in the future, focusing on the pharmacokinetics and pharmacodynamics of ETP.

## 2. Preliminary Study Using Reduced-Dose ETP/CY/TBI as a Conditioning Regimen in Allogeneic Bone Marrow Transplantation for Adults with Hematological Malignancies

The allogeneic bone marrow transplantation (allo BMT) for patients with advanced hematological malignancies including advanced leukemias using the high-dose (usually 60 to 70 mg/kg in a single dose) ETP/TBI conditioning regimens was reported by Blume et al. [14] in 1987 using a 25 to 70 mg/kg ETP dose and by Schmitz et al. [19] in 1988 using a 60 to 70 mg/kg ETP dose. These studies showed some efficacy, and the maximum tolerated dose was shown to be 60 mg/kg [14]; however, severe toxicities were observed. When a single dose of 30 mg/kg ETP was administered, a peak plasma concentration (Cmax) of approximately 100 μg/mL and a trough plasma concentration (Cmin) of approximately 2 to 10 μg/mL were observed [14]. Therefore, we hypothesized that a lowered ETP dose in the divided administration but not in a single dose would be better for avoiding severe toxicities, and we thus decided to add 10 to 25 mg/kg of ETP for two days to the standard CY/TBI conditioning regimen in 1990. Since then, we have routinely used a reduced-dose (10 to 25mg/kg for two days) ETP, 120mg/kg (60 mg/kg for two days) CY, and 12 Gy (six fractions for three days) TBI conditioning regimen in allo BMT for almost all hematological malignancies without any intentional selection and restriction to ALL patients in order to reduce the relapse rate by intensifying the conditioning regimen used for patients at the Hokkaido University Hospital [25]. In the initial dose finding study with the intravenous administration over three hours at a dose of 10 to 25 mg/kg of ETP once daily for two days, the administration of 20 to 25 mg/kg of this agent for two days appeared to cause clinical features resembling hyperacute GVHD associated with elevated serum interleukin-6 levels in three acute leukemia patients who received bone marrow grafts from the human-leukocyte antigen (HLA)-matched sibling donors [48]. Based on this finding, we decided to use the dose of 15 mg/kg for two days as the tentative maximum tolerated dose in 1993. Several reports on the ETP-based conditioning regimens using higher doses (usually 40 to 60 mg/kg as a single dose) were published from 1992 to 2004 [15,16,17,18,19,20,21]; however, severe transplant-related complications were observed, indicating that a reduced dose and divided administration appears to be necessary to avoid the severe adverse events, which were extensively described in our previous review [24]. Basically, the medium-dose ETP/CY/TBI conditioning regimen is performed in the order of ETP (on days -seven and -six), CY (on days -five and -four), and TBI (on days -three, -two, and -one). Although the timing of the ETP administration can be changed to the other days, the synergistic interaction with CY appears to exist in the order of the ETP followed by CY, which exerts a more potent anti-leukemic effect [24]. Therefore, their administration order may be important.

We retrospectively analyzed 38 patients aged 16–49 years (median age: 29 years) who underwent allo BMT for the treatment of hematological malignancies at the Hokkaido University Hospital during the period from May 1990 to December 2002 [25]. Of these, 12, 2, 22, and 2 patients underwent allo BMT from HLA-identical related donors, HLA one antigen-mismatched related donors, HLA-identical unrelated donors, and HLA one antigen-mismatched unrelated donors, respectively. At transplantation, 21 patients were in non-CR. All the patients received medium-dose ETP (intravenous administration over three hours at a dose of 15 mg/kg once daily for two days), CY (60mg/kg for two days) and TBI (12 Gy in six fractions for three days) as a conditioning regimen for allo BMT. Two patients died on day 30 after transplantation. The median follow-up period for all patients was 35.0 months (range: 0.8–159.6 months). At the time of analysis, 10 patients died and seven of those died because of relapse. The estimated 5-year (5-Y) leukemia-free survival (LFS) rates for all, AML, and ALL cases were 73.6%, 66.7%, and 100%, respectively (Table 1). The estimated 5-Y LFS rates for CR1, CR2 and non-CR cases were 90.5%, 83.3%, and 40.9%, respectively (*p* < 0.05). Thus, we decided to use this regimen for only ALL patients rather than for patients with AML or other hematological malignancies at the Hokkaido University Hospital beginning in 2004.

## 3. Addition of Medium-Dose ETP to a CY/TBI Conditioning Regimen in Allo HSCT for Adults with ALL

Almost five years after the first report regarding the efficacy of the medium-dose ETP/CY/TBI in adults with ALL who underwent allo BMT, we retrospectively evaluated the outcome of a consecutive series of 37 adult ALL patients who underwent allo HSCT conditioned with such a regimen during the period from 1993 to 2007 at the Hokkaido University Hospital [26]. We used this regimen for all adults with ALL without any intentional selection. The median age of the patients was 26 years (range: 15–58 years). Of these patients, 13, 18, and six patients underwent transplantation from HLA-matched related donors, HLA-matched unrelated donors, and HLA-mismatched donors, respectively; 32 patients received bone marrow; four patients received peripheral blood stem cells; 10 patients were Ph-positive; and 35 patients were in CR at transplantation. All of the patients achieved engraftment, and grade 3 organ toxicity before engraftment occurred in 27 patients. The incidence rates of the stomatitis, diarrhea, and a febrile episode before engraftment were 57.6%, 33.3%, and 69.7%, respectively. Grade II-III acute GVHD and chronic GVHD occurred in 15 and 18 patients, respectively. No patient developed grade IV acute GVHD or died of GVHD. At a median follow-up of 35.1 months, 32 patients were alive and all Ph -positive patients who were not treated with tyrosine kinase inhibitors either pre- or post- allo HSCT were alive. Three patients died of relapse and two died of TRM. The actuarial 3-Y overall survival (OS), relapse, and TRM rates were 89.2%, 8.1%, and 5.4%, respectively (Table 1). Non-CR at transplantation, MRD, and no acute GVHD were significant adverse prognostic factors for survival when determined by the univariate analysis because of the small numbers of patients. A medium-dose ETP/CY/TBI conditioning regimen for adults with ALL was associated with a lower relapse rate and no increase in toxicity, resulting in better survival compared to the other MAC regimens such as CY/TBI, higher-dose ETP-based regimens, and busulfan-based regimens [24]. No increased second malignancy by using ETP was observed at this time point.

## 4. Retrospective Comparison of the Outcomes between Medium-Dose ETP/CY/TBI and CY/TBI Conditioning Regimens in Adults with ALL who Underwent Allo HSCT

We retrospectively compared the outcomes of the medium-dose ETP/CY/TBI (*n* = 35) and CY/TBI (*n* = 494) conditioning regimens in adults with ALL who underwent allo HSCT [27]. Data for patients who received the medium-dose ETP/CY/TBI conditioning regimen were collected from six centers in Hokkaido and data for patients who received a CY/TBI conditioning regimen were collected from the Japan Society for Hematopoietic Cell Transplantation database (Transplant Registry Unified Management Program) and the Japan Marrow Donor Program database. The enrolled patients met all of the following criteria: First HSCT performed between 1993 and 2007, aged 15–59 years, CR1 or CR2 at allo HSCT, bone marrow or peripheral blood as the stem cell source, HLA-phenotypically 6 loci matched (A, B, and DR loci) related donor or unrelated donor. The median age of the patients was 34 years (range: 15–59 years), and patients who received the medium-dose ETP/CY/TBI were younger (28 vs 34 years, *p* = 0.02). The cumulative incidences of relapse and non-relapse mortality (NRM) were higher for patients who received CY/TBI (*p* = 0.01 for relapse, *p* < 0.01 for NRM) (Table 2). After a median follow-up period of 36.9 months, the 5-Y OS rates were 82.2% in the medium-dose ETP/CY/TBI conditioning regimen and 55.2% in the CY/TBI conditioning regimen. The OS and LFS in the medium-dose ETP/CY/TBI conditioning regimen were shown to be significantly better by the multivariate analysis for LFS (hazard ratio (HR): 0.21; 95% confidence interval (CI): 0.06-0.49; *p* < 0.01) and for OS (HR: 0.25; 95% CI: 0.08-0.59; *p* < 0.01) (Table 2). The medium-dose ETP/CY/TBI conditioning regimen was associated with a lower relapse rate and no increase in NRM, resulting in better survival than in the CY/TBI conditioning regimen for adults with ALL. There were no significant differences between the two regimens in terms of engraftment, or incidences of acute and chronic GVHD.

## 5. A prospective Multi-Center Phase II Clinical Trial with a Medium-Dose ETP/CY/TBI Conditioning Regimen for Adults with ALL in Japan

To prospectively analyze the efficacy of a medium-dose ETP/CY/TBI conditioning regimen for adults with ALL, we conducted a prospective multi-center phase II clinical trial in Japan from 2009 to 2011, followed by a 2-year follow-up [28]. The eligibility criteria of this study were as follows: Diagnosis of ALL or acute biphenotypic leukemia, defined by the criteria proposed by the European Group for the Immunological Classification of Leukemias; age between 15 and 50 years; in hematological CR; Eastern Cooperative Oncology Group performance status of 0 to two; adequate functions of main organs, including the liver, kidneys, lungs, and heart; and HLA serologically A, B, DR, 6 of 6 matched related or unrelated donor. The stem cell source was limited to the bone marrow or peripheral blood. Patients who received the previous autologous or allo-HSCT, and those with Burkitt leukemia were ineligible for this study. Between February 2009 and August 2011, 52 patients were enrolled from 19 transplant centers in Japan, but two patients who underwent allo HSCT from HLA serologically mismatched donors were excluded from the analysis. The median age of the patients was 33.5 years (range: 17–49 years). Nineteen (42.0%) of the patients were Ph-positive, and all of the Ph-positive patients received tyrosine kinase inhibitors (imatinib, *n* = 15; dasatinib, *n* = 4) as a part of the pre-transplant treatment. Forty seven (94.0%) of the patients were in CR1 at transplantation, and three (6.0%) were in CR2. Twenty six (52.0%) of the patients underwent allo HSCT from an HLA-A-, -B-, -DR-matched related donor and 24 (48.0%) underwent allo HSCT from an HLA-A, -B-, -DR-matched unrelated donor. Forty (80.0%) of the patients received bone marrow, and 10 (20.0%) received peripheral blood stem cells (PBSC). The PBSC were from a related donor in all cases because the donation of PBSC from unrelated donors was not permitted during this study in Japan. All patients achieved the neutrophil engraftment. The incidence rates of stomatitis, diarrhea, nausea/vomiting, and a febrile episode before engraftment were 60.0%. 26.0%, 38.0%, and 72.0%, respectively. Grade III-IV acute GVHD and extensive chronic GVHD developed in four and 18 patients, respectively. No patient died within 100 days after allo HSCT. The 1-Y event-free survival (EFS) was 76.0%, and the 1-Y OS was 80.0% (Table 1). The cumulative incidences of relapse and NRM at one year after allo HSCT were 10.0% and 14.0%, respectively.

This prospective multi-center phase II clinical trial in Japan clearly showed that the medium-dose ETP/CY/TBI conditioning regimen is efficacious in adults with ALL, similarly to the previous results [25,26,27].

## 6. Recent Outcomes of Medium-Dose ETP/CT/TBI and CY/TBI Conditioning Regimens in Allo HSCT for Adults with ALL in Japan

A retrospective cohort study was conducted using a Japanese transplant registry database to compare the prognosis between the medium-dose ETP/CY/TBI (ETP, total 30 to 40 mg/kg) (*n* = 376) and CY/TBI (*n* = 1178) conditioning regimens in adults with ALL transplanted at CR between January 1 2000 and December 31 2014 [29]. We are pleased with the fact that more than 300 adults with ALL underwent allo HSCT with the medium-dose ETP/CY/TBI conditioning regimen in the almost three years after enrollment in the prospective multi-center phase II clinical trial in Japan was completed. The medium-dose ETP/CY/TBI conditioning regimen significantly reduced relapse compared with the CY/TBI conditioning regimen (1-Y LFS: 71.5% and 3-Y LFS: 59.6%; HR: 0.75; 95% CI: 0.56–1.00; *p* = 0.05) with a corresponding improvement in LFS (1-Y LFS: 76.0% and 3-Y LFS: 65.5%; HR: 0.76; 95% CI: 0.62–0.93; *p* = 0.01), particularly in patients transplanted at CR1 with advanced-risk (positive minimal residual disease, presence of poor-risk cytogenetics, or an initial elevated leukocyte count) (HR: 0.75; 95% CI: 0.56–1.00; *p* = 0.05) or those transplanted beyond CR2 (HR: 0.58; 95% CI: 0.39–0.88; *p* = 0.01) (Table 2). The addition of ETP did not increase the incidence of post-transplant complications or NRM (HR: 0.88; 95% CI: 0.65–1.18; *p* = 0.38).

This study again confirmed our previous results [25,26,27,28]. The tendency to a slightly higher TRM rate in this study may be due to the fact that data from patients who received 40 mg/kg ETP were included as well as data from those who received 30 mg/kg ETP.

## 7. Pharmacological Study of ETP in a Medium-Dose ETP/CY/TBI Conditioning Regimen before Allo HSCT

Although the pharmacokinetics and pharmacodynamics (PK/PD) of ETP in patients with various malignancies have been extensively analyzed so far, there exists only one study in two patients with leukemia who received the high-dose ETP-based conditioning regimens in allo BMT [14]. Tazawa et al. [49] investigated the pharmacokinetics of ETP to reduce the inter-individual variations of the ETP plasma concentrations in 20 adults with acute leukemia who underwent allo HSCT with a medium-dose ETP/CY/TBI conditioning regimen between April 2008 and January 2013 at the Hokkaido University Hospital. The peak concentration (Cmax) and the area under the plasma concentration-time curve (AUC) of ETP differed greatly among patients (range of Cmax, 51.8 to 116.5 μg/mL; range of AUC, 870 to 2015 μg·h/mL). There was a significant relationship between Cmax and AUC (R = 0.85, *p* < 0.05). The distribution volume (Vd) was one of the factors of the inter-individual variation in the plasma concentration of ETP in each patient (range of Vd, 0.13–0.27 L/kg), correlating with the albumin and body weight (R = 0.56, *p* < 0.05; R = 0.40, *p* < 0.05 respectively). A plasma ETP concentration more than 75.6 μg/ mL was associated with a high mortality rate.

ETP is a topoisomerase II inhibitor that suppresses the cell cycle progression at a premitotic phase (late S and G2), via inhibition of DNA synthesis [50,51,52,53,54,55,56,57]. Therefore, the schedule dependency in both animal models and a clinical trial has been observed, indicating that multiple dosing over three to five consecutive days is superior to a weekly single dose administration due to the cell-cycle specific (and, thereby, time-dependent) anti-tumor activity [58,59].

In fact, Slevin et al. [60] showed that patients with the small-cell lung cancer (SCLC) receiving 500 mg/m^2^ ETP either as a 24-h intravenous infusion or a daily 2-h infusion for five days have positive response rates of 10% or 89%, respectively. Although there was no significant difference in total AUC measurements between these two treatment arms, prolonged maintenance of low plasma ETP concentrations (≥1 μg/mL) was associated with superior efficacy in the 5-day treatment arm. In a subsequent study, with SCLC, five- and eight- day schedules were found to have equivalent anti-tumor activity [61]. Hematological toxicities were greater in the five-day treatment arm, suggesting that prolonged exposure to low concentrations of ETP appears to improve its efficacy. A similar finding was reported in patients with Hodgkin lymphoma and SCLC receiving 25 mg/m^2^/day [62]. A mean plasma ETP concentration of 0.78 ± 0.4 μg/ mL, (roughly 1 μg/mL) is required to maintain the anti-tumor activity, without severe myelotoxicity. Another investigation showed that the plasma ETP concentration of 0.75 to 1.0 μg/mL is sufficient to exert anti-tumor activity in patients with SCLC and in patients with non-SCLC, a plasma ETP concentration of 1.0 to 2.0 μg/mL is required to induce a positive response [63]. These findings showed that the prolonged maintenance of a low plasma ETP concentration (approximately 1 μg/mL) was associated with superior anti-tumor activity with minimal toxicities, while the optimal dose slightly differed depending on the type of malignancy.

In the medium-dose ETP/CY/TBI conditioning regimen in which intravenous administration over three hours at a dose of 15 mg/kg of ETP once daily for two days was performed, the Cmin (day one: 3.9 ± 2.5 μg/mL, median: 3.7 μg/mL; day two: 6.2 ± 3.7 μg/mL, median: 5.4 μg/mL) readily exceeds such a plasma concentration (1 μg/mL) [49]. To maintain a plasma concentration of 1 μg/mL as the Cmin, a 1/5 to 1/3 dose of ETP (namely, 3 mg/kg to 5 mg/kg for two days but not 15 mg/kg for two days) may be enough. Furthermore, since more prolonged administration is recommended to obtain a better outcome without increasing toxicities, the ETP administration should be spread out to three to four days rather than two days. Thus, it is meaningful to verify whether less than 15 mg/kg/day ETP added to a CY/TBI conditioning regimen is feasible or not and whether more prolonged administration of ETP more than two days is also appropriate or not. Since ETP is known to possess various immunological functions that may play a role in eliminating leukemic cells [24] in addition to the proper anti-tumor activity, an extensively lowered dose may be efficacious. When it is proven that a more reduced-dose of ETP will exert a positive response, we could use this drug in addition to the presently available various RIC regimens as a new RIC regimen.

To answer these questions, a prospective phase III clinical trial is awaited to compare the medium-dose ETP/CY/TBI and CY/TBI conditioning regimens. However, in an attempt to conduct a prospective phase III clinical trial, there is one obstacle to be overcome: namely, the probable difficulty of enrolling sufficiently many adults with ALL because of the presence of many treatment modalities other than allo HSCT for such patients at the present.

A PK/PD study of ETP is important to determine an optimal dose for each patient in order to minimize adverse events and maximize efficacy; however, the plasma concentration is not always a good surrogate for the pharmacologically active intratumoral (intracellular) concentration [64,65,66], probably due to the variable drug transporter function [67,68]. Therefore, we are required to carry out a PK/PD study to find an optimal dose for each patient by analyzing both the plasma and intracellular concentrations of the relevant drug over time.

## 8. Conclusions

A medium-dose ETP/CY/TBI conditioning regimen appears to be useful for adults with ALL, showing a better outcome with a lower incidence of relapse and TRM. The medium-dose ETP/CY/TBI conditioning regimen was found to be more efficacious in higher risk ALL patients; therefore, allo HSCT appears to be suitable for advanced-risk patients at CR1 and standard-risk patients at CR2 (probably CR1 as well). A recommended ETP dose is 15 mg/kg for two days or less to decrease the TRM and relapse rates, preserving a better outcome. PK/PD studies of both plasma and intracellular concentrations will be helpful to determine an optimal ETP dose for each patient.

## Figures and Tables

**Table 1 jcm-08-00569-t001:** The efficacy and safety of the medium-dose ETP/CY/TBI conditioning regimen in allogeneic hematopoietic stem cell transplantation for patients with adult acute lymphoblastic leukemia.

Study Design	No. of Patients	Overall Survival	Leukemia-Free Survival	Relapse Rate	Non-Relapse Mortality	Reference
Retrospective	11	100% (5-years)	100% (5-years)	0%	0%	[25]
Retrospective	37	89% (3-years)		8%	5%	[26]
Prospective phase II	50	80% (1-years)	76% (1-years)	10%	14%	[28]
		67% (2-years)	65% (2-years)			

**Table 2 jcm-08-00569-t002:** Comparison of outcomes between the medium-dose ETP/CY/TBI and CY/TBI conditioning regimens in allogeneic hematopoietic stem cell transplantation for patients with adult acute lymphoblastic leukemia.

Regimen	No. of Patients	ETP Dose (mg/kg)	Overall Survival		leukemia-Free Survival		Relapse Rate	Non-Relapse Mortality	Reference
			2 years	5 years	2 years	5 years			
ETP/CY/TBI	35	30	91%	82%	90% *	82% *	14% *	3%	[27]
CY/TBI	494		68%	55%	55% *	50% *	29%	7%	
			1 year	3 years	1 year	3 years			
ETP/CY/TBI	376	30–40	85% *	75% *	76%	66%	20% *	17%	[29]
CY/TBI	1178		75% *	65% *	72%	60%	25% *	18%	

* Estimated according to the corresponding figure.

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
