# Peer review of "Perspectives on the Use of a Medium-Dose Etoposide, Cyclophosphamide, and Total Body Irradiation Conditioning Regimen in Allogeneic Hematopoietic Stem Cell Transplantation: The Japanese Experience from 1993 to Present"

_jcm, 2019, doi:10.3390/jcm8050569_

Reviewer 1 Report

This work is a summary of the experience with intermediate dose etoposide, cyclophosphamide, and TBI in Japan as a conditioning regimen for transplantation

Comments:

  The authors make statements such as "RIC is suitable for patient aged over 50 years" but no data is presented to support this statement..

Because serial studies are reported upon here, there seems to be overlap in years, and therefore one would presume that some patients were included in more than one analysis.

The dose selection for VP-16 sounds quite empiric (top of page 3);e.g., "we supposed that a dose of 10 mg/kg or less might also be efficacious."

The formatting in Table 1 seems off.

The inclusion of multiple disease types un each serial study detracts from the whole.

In section 3, page 5, it is not described how prognostic variables were analysed (was this a multivariable analysis)?

Statements such as "was associated with lower relapse rate and non increased in toxicity, resulting in better survival" need to be justified; compared to what, for example?

At the top of page 7, the authros describe HLA-Cw matching, but they do not follow up on this.

In section 6, confidence intervals are used; % survival with CIs would be more helpful.

At the bottom of page 7, there is a statement about rats. It is not certain how this relates to the human studies described.

The first sentence of the conclusions seems overstated since no randomized study of CY/TBI for medium-dose ETP/Cy/TBI has been completed. Caution in overstating is advised.

The title should be changed as not all the studies cited were restricted to ALL. Maybe just change it to "Perspectives on the Use of medium-dose etoposide, cyclophosphamide, and total body irradiation; the Japanese experience from 1993 to present.

Some style and grammar changes are required throughout.

Author Response

Reply to the reviewer 1

Thank you very much for your constructive and helpful suggestions. We edited our manuscript according to your comments.

1.     The authors make statements such as “RIC is suitable for patient aged over 50 years” but no data is presented to support this statement.

We added a paper entitled “Reduced-intensity vs myeloablative conditioning allogeneic hematopoietic SCT for patients aged over 45 years with ALL in remission: a study from the Adult ALL Working Group of the Japan Society for Hematopoietic Cell Transplantation (JSHCT) reported by Tanaka et al. which was published in Bone Marrow Transplant. (2013, 48, 1389-1394). They showed that RIC is preferable to patients aged over 55 years.  So, we changed the limiting age for RIC from 50 years to 55 years.

2.     Because serial studies are reported upon here, there seems to be overlap in years, and therefore one would presume that some patients were included in more than one analysis.

As you pointed out, some patients were overlapped in serial retrospective studies. But, it is inevitable because the numbers of patients is increasing year by year. In each studies, similar outcome were constantly observed by the medium-dose ETP/CY/TBI conditioning regimen.

3.     The dose selection for VP-16 sounds quite empiric (top of page 3);e.g., "we supposed

that a dose of 10 mg/kg or less might also be efficacious."

Although the initial dose selection for ETP was empiric, the decision was made

by considering the findings available at that time.  Therefore, it is not so unreasonable. In general, the optimal dose of anti-cancerous agents in Western countries appears to be more toxic for Japanese. Furthermore, the previous reports using 25-70 mg/kg in a single dose suggested that higher dose ETP was too toxic. So, we preferred to use 10-25 mg/kg for 2 days; however, 20-25 mg/kg for 2 days induced features resembling hyperacute graft versus host disease. Thus, we decided to use 15 mg/kg for 2 days at that time. Although the pharmacological study by Tazawa et al. (Pharmacokinetics and dose adjustment of etoposide administered in a medium-dose etoposide, cyclophosphamide and total body irradiation regimen before allogeneic

hematopoietic stem cell transplantation. J. Pharm. Health Care Sci. 2016, 2, 18.)

suggests that 10 mg/kg or less may be enough, we deleted the following sentence “we

supposed that a dose of 10 mg/kg or less might also be efficacious”.

4.     The formatting in Table 1 seems off.

According to your comment, we deleted Table 1.

5.     The inclusion of multiple disease types in each serial study detracts from the whole.

      We deleted the description on multiple disease types.

6.     In section 3, page 5, it is not described how prognostic variables were analysed (was this a multivariable analysis)?

      The prognostic variable was determined by univariate analysis, because the numbers of patients were so small for multivariate analysis.  This was described in the text.

7.     Statements such as "was associated with lower relapse rate and non increased in toxicity, resulting in better survival" need to be justified; compared to what, for example?

We added the sentence that “compared to CY/TBI, higher-dose ETP-based, and

busulfan-based conditioning regimens”.

8.     At the top of page 7, the authors describe HLA-Cw matching, but they do not follow up on this.

We deleted the description on HLA-Cw matching.

9.     In section 6, confidence intervals are used; % survival with CIs would be more helpful.

We added the description of % survival.

10.   At the bottom of page 7, there is a statement about rats. It is not certain how this relates to the human studies described.

We deleted the description on experiment using rats.

11.  The first sentence of the conclusions seems overstated since no randomized study of CY/TBI for medium-dose ETP/Cy/TBI has been completed. Caution in overstating is advised.

We changed the first sentence of the conclusions to milder description.

12.  The title should be changed as not all the studies cited were restricted to ALL. Maybe just change it to "Perspectives on the Use of medium-dose etoposide, cyclophosphamide, and total body irradiation; the Japanese experience from 1993 to present.

We changed the title according to your suggestion as follows: “Perspectives on the use of medium-dose etoposide, cyclophosphamide, and total body irradiation conditioning regimen in allogeneic hematopoietic stem cell transplantation: the Japanese experience from 1993 to present”.

13.   Some style and grammar changes are required throughout.

MDPI English editing was carried out.

Reviewer 2 Report

Adult acute lymphoblastic leukemia is a relatively poor outcome disease. Allogeneic hematopoietic stem cell transplantation may improve it's outcome. This authors just reviewed their retrospective and prospective data in the use of medium-dose etoposide, cyclophosphamide,and TBI conditioning regimen in their patients. Although, the manuscript had been well organized  and presented their data with interesting results in the comparing the results of conventional conditioning regimen with CY/TBI. Some small comments was recommended.

The order and layout of Table I, Table 2, Table 3 are in missequence. They need to be checked.

In Table 2 and Table 3, the authors used leukemia free survival rate in their data, but they used disease free survival in their manuscript (Page 3, Line 132-134. For the disease free survival in leukemic patients, the terminology of leukemia free survival may be more appropriate.

In this manuscript, the authors described the pharmacological study on ETP in their medium-dose conditioning regimen and found the Cmin day 1:3.9+/-2.5ug/mL, median:3.7ug/mL; day 2 6.2+/-3.7ug/mL, median:5.4ug/mL exceeds the plasma concentration of 1ug/mL. However, there were no data to touch the side effects of ETP related to this drug level. The drug toxicity should be related to the drug level. In this manuscript, the authors better present the information of side effects in their medium- dose of ETP. 

Author Response

Reply to the reviewer 2

Thank you very much for your constructive and helpful suggestions. We edited our manuscript according to your comments.

1.     The order and layout of Table 1, Table 2, Table 3 are in missequence. They need to be checked.

We checked Table sequence.

2.     In Table 2 and Table 3, the authors used leukemia free survival rate in their data, but they used disease free survival in their manuscript (Page 3, Line 132-134. For the disease free survival in leukemic patients, the terminology of leukemia free survival may be more appropriate.

We changed the terminology of disease-free survival to that of leukemia-free survival.

3.     In this manuscript, the authors described the pharmacological study on ETP in their medium-dose conditioning regimen and found the Cmin day 1:3.9+/-2.5ug/mL, median:3.7ug/mL; day 2 6.2+/-3.7ug/mL, median:5.4ug/mL exceeds the plasma concentration of 1ug/mL. However, there were no data to touch the side effects of ETP related to this drug level. The drug toxicity should be related to the drug level. In this manuscript, the authors better present the information of side effects in their medium- dose of ETP.

We added the description of the side effects that were frequently observed.

Round  2

Reviewer 1 Report

None.